

# Variability within a clonal population of *Erwinia amylovora* disclosed by phenotypic analysis

Rafael J. Mendes[1,2,3,4,5], Conceição Amaro[6], João Pedro Luz[6], Fernando Tavares[1,4,5] and Conceição Santos[1,2]

[1] Biology Department, Faculty of Sciences, University of Porto, Porto, Portugal
[2] LAQV-REQUIMTE, Faculty of Sciences, University of Porto, Porto, Portugal
[3] CITAB, University of Trás-os-Montes e Alto Douro, Vila Real, Portugal
[4] CIBIO, Research Centre in Biodiversity and Genetic Resources, InBIO, Associated Laboratory, Campus Agrário de Vairão, Faculty of Sciences, University of Porto, Vairão, Portugal
[5] BIOPOLIS Program in Genomics, Biodiversity and Land Planning, CIBIO, Campus de Vairão, Vairão, Portugal
[6] QRural, School of Agriculture, Polytechnic Institute of Castelo Branco, Castelo Branco, Portugal

## ABSTRACT

**Background**. Fire blight is a destructive disease of pome trees, caused by *Erwinia amylovora*, leading to high losses of chain-of-values fruits. Major outbreaks were registered between 2010 and 2017 in Portugal, and the first molecular epidemiological characterization of those isolates disclosed a clonal population with different levels of virulence and susceptibility to antimicrobial peptides.

**Methods**. This work aimed to further disclose the genetic characterization and unveil the phenotypic diversity of this *E. amylovora* population, resorting to MLSA, growth kinetics, biochemical characterization, and antibiotic susceptibility.

**Results**. While MLSA further confirmed the genetic clonality of those isolates, several phenotypic differences were recorded regarding their growth, carbon sources preferences, and chemical susceptibility to several antibiotics, disclosing a heterogeneous population. Principal component analysis regarding the phenotypic traits allows to separate the strains Ea 630 and Ea 680 from the remaining.

**Discussion**. Regardless the genetic clonality of these *E. amylovora* strains isolated from fire blight outbreaks, the phenotypic characterization evidenced a population diversity beyond the genotype clonality inferred by MLSA and CRISPR, suggesting that distinct sources or environmental adaptations of this pathogen may have occurred.

**Conclusion**. Attending the characteristic clonality of *E. amylovora* species, the data gathered here emphasizes the importance of phenotypic assessment of *E. amylovora* isolates to better understand their epidemiological behavior, namely by improving source tracking, make risk assessment analysis, and determine strain-specific environmental adaptations, that might ultimately lead to prevent new outbreaks.

Corresponding author
Rafael J. Mendes,
rafael.mendes@fc.up.pt

## INTRODUCTION

The infectious disease fire blight affects highly valuable pome fruit trees, namely, apple (*Malus domestica*), pear (*Pyrus communis*), quince (*Cydonia oblonga*), and loquat (*Eriobotrya japonica*) that belong to the Rosacea family (*Vanneste, 2000*), and has also been observed in many other ornamental and wild species (*Sundin, 2014*; *Marco-Noales et al., 2017*). The etiological agent responsible for this disease is the Erwiniaceae Gram-negative *Erwinia amylovora* (*Vanneste, 2000*; *Adeolu et al., 2016*), a plant pathogen recommended as quarantine pests by EPPO (*EPPO, 2020*), characterized by a high aggressiveness paralleled by a lack of efficient phytosanitary treatments. Described in the late 1700s in North America, it was only in the second half of the XX$^{th}$ century that *E. amylovora* reached many other countries (*Llop et al., 2008*; *McGhee & Sundin, 2012*; *Pucci, L'Aurora & Loreti, 2013*; *Llorente et al., 2017*; *Song et al., 2020*; *Mendes et al., 2021a*). Intensive agriculture and the free-trade agreement among countries have facilitated the distribution of susceptible cultivars to fire blight, making commercial exchange an important factor in disease dissemination (*Ivanović et al., 2012*).

To understand this dissemination some studies genetically characterized the *E. amylovora* strains epidemiology at local, national, and regional levels (*Halupecki et al., 2006*; *Végh et al., 2012*; *Rhouma et al., 2014*; *Doolotkeldieva et al., 2019*; *Popović et al., 2020*). In particular, the multi locus sequencing analysis (MLSA) provided phylogenetic studies of *E. amylovora* (*Facey et al., 2015*; *Doolotkeldieva et al., 2019*), and of other phytopathogenic bacteria (*Wicker et al., 2012*; *Flores et al., 2018*; *Martins et al., 2020*). Other molecular genotyping techniques were used to characterize this species, such as clustered regularly interspaced short palindromic repeat (CRISPR) sequences, variable number of tandem repeats (VNTR) and multiple-locus variable number tandem repeat analysis (MLVA). Overall, these techniques have disclosed *E. amylovora* as globally highly homogeneous (*Rezzonico, Smits & Duff, 2011*; *McGhee & Sundin, 2012*; *Alnaasan et al., 2017*; *Song et al., 2020*; *Mendes et al., 2021a*). Few differences among populations/strains, were observed mostly in North America, and among Rubus-infecting strains (*Rezzonico, Smits & Duff, 2011*; *McGhee & Sundin, 2012*; *Bühlmann et al., 2014*; *Tancos & Cox, 2016*).

This low genomic diversity is common in contagious epidemiological pattern species, and thus other characterization approaches such as those based on phenotypic traits (*Maruţescu, Sesan & Manole, 2008*; *Gaganidze et al., 2021*) may be valuable in providing better information regarding local environmental adaptative pressure. Additionally, in a previous study, it was demonstrated that phenotypic variation occurs in highly homogeneous *E. amylovora* strains (*Zeng et al., 2018*). Biochemical traits have been used to characterize *E. amylovora*, resorting to techniques such as the bioMérieux API 20E system and the Biolog Microbial Identification System (*Maruţescu, Sesan & Manole, 2008*; *Constantinescu et al., 2011*; *Myung et al., 2016*). Studies using API 20E showed a high homogeneity for *E. amylovora* strains (*Maruţescu, Sesan & Manole, 2008*; *Constantinescu et al., 2011*), whilst the Biolog system allowed to disclose distinct groups of *E. amylovora* in a population based on carbon source utilization, since it allowed to disclose strains

possessing higher metabolic activity due to metabolization of additional carbon sources (*Ivanović et al., 2012*).

Due to inefficient control methods against aggressive phytopathogens like *E. amylovora*, antibiotics-based management (re)emerged in some countries (*Sundin & Wang, 2018*). Streptomycin and oxytetracycline are the main antibiotics against *E. amylovora* (*McManus et al., 2002*), and gentamicin and oxolinic acid have been applied to control fire blight in Mexico and Israel, respectively (*Shtienberg et al., 2015*). However, public awareness and the finding of strains with antibiotic resistance have led several countries, including the European Union, to restrict their use (*Stockwell & Duffy, 2012*; *Sundin & Wang, 2018*). In the USA, streptomycin can be used against *E. amylovora*, while oxytetracycline may be used in regions, where streptomycin-resistant strains appeared (*Stockwell & Duffy, 2012*). Similarly, although rare, countries using Cu-based agrochemicals face the appearance of Cu-resistant *E. amylovora* populations (*La Torre, Iovino & Caradonia, 2018*). Thus, antibiotic susceptibility is an important tool to disclose variability of bacterial populations.

First fire blight outbreaks appeared in 2010 in Portugal, being persistent and with periodic outbreaks, which cause dramatic economic impacts mostly to apple and pear producers (*EPPO, 2018*; *Mendes et al., 2021a*; *Mendes et al., 2021b*). A study from *Kurz et al. (2021)* that includes six strains from *E. amylovora* isolates from 2010 and 2011 has shown low diversity regarding those strains, with only two CRISPR profiles disclosed, similar to our previous findings (*Mendes et al., 2021a*), namely a CRISPR profiling of Portuguese representative temporal/local populations of *E. amylovora* associated with outbreaks occurring between 2010 and 2017, which displayed two CRISPR profiles. Nevertheless, it was also revealed that despite the high clonality of the studied *E. amylovora* isolates, distinct levels of virulence were observed (*Mendes et al., 2021a*).

In this study, we assess growth kinetics, biochemical diversity (carbon source and chemical sensitivity), antibiotics susceptibility, as core phenotypic traits capable to characterize environmental isolates, and disclose epidemiological patterns of *E. amylovora* strains within a genetically clonal Portuguese population responsible for several fire blight outbreaks occurring in pome fruits orchards. For that, several representative field isolates were characterized by MLSA, BIOLOG profile, and growth curves. Also, strains' antibiograms for several antibiotics were profiled, to disclose populations' variability against them. Finally, to correlate the results obtained, a multivariate analysis was performed. Altogether the data contributes to better characterize *E. amylovora* strains associated to major fire blight outbreaks occurring in Portugal, and provides new insights into its dissemination and entry routes.

## MATERIALS & METHODS

### Bacterial strains and culture conditions

The strains used in this study integrate a Portuguese population, previously characterized regarding genetic markers and susceptibility to antimicrobial peptides (AMPs) (*Mendes et al., 2021a*; *Mendes et al., 2021b*). This population included 36 Portuguese *E. amylovora* strains, that disclosed a new CRISPR profile for one strain (Ea 680), different virulence

**Table 1** *Erwinia amylovora* strains used.

| Strain | Host | | Isolated from | Geographic origin | Year |
|---|---|---|---|---|---|
| | **Species** | **Cultivar** | | | |
| Ea 230 | Pear | 'Rocha' | Exudate | Alcobaça, Portugal | 2010 |
| Ea 320 | Pear | 'Rocha' | Branch | Alcobaça, Portugal | 2011 |
| Ea 390 | Apple | 'Royal Gala' | Necrotic fruit | Alcobaça, Portugal | 2011 |
| Ea 490 | Pear | 'Rocha' | Branch | Alenquer, Portugal | 2015 |
| Ea 630 | Apple | 'Gala' | Branch | Cadaval, Portugal | 2015 |
| Ea 680 | Pear | 'Rocha' | Branch | Cadaval, Portugal | 2015 |
| Ea 820 | Pear | Unidentified | Branch | West[a], Portugal | 2017 |
| CFBP 1430 | *Crataegus* | – | Unidentified | Nord, France | 1972 |
| LMG 2014 | Pear | Unidentified | Unidentified | United Kingdom | 1959 |

**Notes.**
[a]This strain have been isolated in the West region of Portugal, which includes the municipalities Alcobaça, Caldas da Rainha, Alenquer and Cadaval.

levels that grouped most of the strains in two levels (4 and 5), with one strain being less virulent (Level 3—Ea 680), and two most virulent (Level 6—Ea 620 and Ea 630), and susceptibility levels between the strains (*Mendes et al., 2021a*; *Mendes et al., 2021b*).

From those studies, the seven most distinctive strains regarding CRISPR, susceptibility to chemicals, and virulence profiles, that were collected from different apple and pear producing regions in 2010, 2011, 2015, and 2017, were selected (Table 1). The reference strain CFBP 1430 and the type strain LMG 2024 were used as references. *E. amylovora* strains were preserved in 70% King's B (KB) medium and 30% glycerol at −80 °C. Throughout this study, strains were cultured at 28 °C in KB medium, unless differently stated.

## Specific growth rate determination

Pure colonies of each strain of *E. amylovora* were grown in KB and incubated during 16 h at 25 °C with 180 rpm. Optical density (OD) was adjusted to 0.1 at 600 nm ($OD_{600}$) for each strain. Growth curves were obtained in a 96-well titration plate containing 100 μL of bacterial culture in each well, in a multiplate reader (Mulstiskan$^{TM}$ Go, Thermo Fisher, USA) at 25 °C with constant shaking for 24 h. $OD_{600}$ was obtained each hour. Three independent assays were performed. Specific growth rate (μ) was determined through the exponential growth phase of each strain.

## MLSA

DNA extraction, followed by amplification of four pair of primers recommended by EPPO to identify *E. amylovora* was performed as described in *Mendes et al. (2021a)*. Four housekeeping genes were employed for MLSA analysis, namely *gyrB*, *gapA*, *rpoD*, and *infB*, which encodes for DNA gyrase subunit B, glyceraldehyde-3-phosphate dehydrogenase A, RNA polymerase sigma factor, and translation initiation factor IF-2, respectively, using the primers listed in Table 2.

Amplification of these genes through polymerase chain reaction (PCR) was carried out using a reaction mixture containing 1× of 2×PCR Master Mix (Bioron, Germany), 0.2 μM of each primer, and 10 ng of DNA template, in a final volume of 20 μL. PCR cycling

**Table 2  List of primers used.**

| Primer | Sequence (5′-3′) | Reference |
| --- | --- | --- |
| *gyrB* F | MGGCGGYAAGTTCGATGACAAYTC | *Sarkar & Guttman (2004)* |
| *gyrB* R | TRATBKCAGTCARACCTTCRCGSGC | *Sarkar & Guttman (2004)* |
| Erw_*infB* F | CGTGATGCCACAGACTATCG | This work |
| Erw_*infB* R | CTTTCACTTCATCAGTAGACAGC | This work |
| Erw_*rpoD* F | GGCGATAGAGATAACCAGACG | This work |
| Erw_*rpoD* R | GCGTGAAATGGGTACGGTTG | This work |
| *gapA*-7 F | ATCAAAGTAGGTATCAACGG | *Cigna et al. (2017)* |
| *gapA*-938 R | TCRTACCARGAAACCAGTT | *Cigna et al. (2017)* |

conditions were carried out with a first amplification cycle of 2 min at 94 °C, followed by 30 cycles at 94 °C for 20 s, 54 °C (*gapA*)/57 °C (*infB*)/59 °C (*rpoD*)/60 °C (*gyrB*) for 20 s, and 72 °C for 60 s, and a final extension at 72 °C for 5 min. PCR amplicons were separated in a 1.2% agarose gel stained with GreenSafe Premium (NZYTech, Portugal), visualized using the GelDoc (Bio-Rad Laboratories, USA) and purified using the illustra GFX$^{TM}$ PCR DNA and Gel Band Purification Kit (GE Healthcare, USA), following manufacturer's instructions. Sequencing was outsourced to STAB Vida (Costa da Caparica, Portugal). Raw sequences were analyzed with Geneious program version 11 (Biomatters, New Zealand). For MLSA other *E. amylovora* sequences available at GenBank were retrieved, alongside with sequences from other species (Table S1). The sequences of each locus were aligned and trimmed resorting to the Geneious program version 11. A dendrogram from four-locus concatenated sequences was generated using neighbor-joining (UPGMA) and 1,000 bootstrap iterations.

## Nucleotide sequences accession numbers

DNA sequences corresponding to the four housekeeping genes (*gyrB*, *gapA*, *rpoD*, and *infB*) of the seven *E. amylovora* Portuguese strains were deposited in the National Center for Biotechnology Information (NCBI) database with the following accession number: MW647223 to MW647229 for *gapA*; MW647230 to MW647236 for *gyrB*; MW647237 to MW647243 for *infB*; and MW647244 to MW647250 for *rpoD* (Tables S2 and S3).

## Carbon source and chemical sensitivity phenotyping

The *E. amylovora* strains were characterized through their biochemical pattern resorting to Biolog GEN III Microplate$^{TM}$ system (Biolog$^{TM}$, USA), according to manufacturer's instructions. Briefly, the strains were first grown on solid YNA medium (4 g of meat extract; 5 g of peptone; 2.5 g yeast extract; 5 g of NaCl; 15 g of agar; distilled water up to 1 L; pH 7.0) for 48 h at 28 °C. Fresh colonies were transferred, with a cotton-tipped swab, to new vials containing Inoculating Fluid A. The inoculum density was adjusted to a transmittance of 95–98% resorting to a turbidimeter. After that, 100 μL of the inoculum was dispensed into each well of the Biolog MicroPlate. MicroPlates were then incubated at 30 °C during 24 h, and then the plates were read in a microplate photometer (Mulstiskan$^{TM}$ FC; Thermo Fisher Scientific, Waltham, MA, USA). For all strains, two independent replicates were

performed in different dates. Results were considered positive if the OD at 595 nm ($OD_{595}$) was higher than 50% of the positive control, whilst they were considered negative if the $OD_{595}$ was below 25% of the positive control. Results between these two parameters were considered borderlines (*Flores et al., 2018*). The dendrogram (UPGMA, bootstrap of 1,000) was obtained resorting to RStudio (*RStudio Team, 2020*).

## Antibiogram assays

The susceptibility of the nine *E. amylovora* strains against four antibiotics was assessed through disc-diffusion antibiotic susceptibility tests. Briefly, bacterial strains were grown for 16 h at 25 °C with 180 rpm in Mueller-Hinton (MH) broth. Next, $OD_{600}$ was adjusted for every strain at 0.1, followed by streaking them into a Petri dish with MH medium with a swab, and one disk for each antibiotic, namely, Streptomycin 10 μg (S10), Oxytetracycline 30 μg (OT30), Gentamicin 10 μg (CN10), and Oxolinic acid 2 μg (OA2) were dispensed into the Petri dish. The bacteria were incubated at 25 °C for 24 h, and photographs were taken for each one with GelDoc (Bio-Rad Laboratories, USA). The areas of the halos obtained were measured resorting to ImageJ (National Institutes of Health, USA). The assay was independently repeated three times.

## Statistical analysis

Comparisons between the strains for the μ and antibiogram assay were analyzed through one-way ANOVA, followed by Tukey's for multiple comparisons test, using GraphPad Prism 9 for Windows (GraphPad Software, San Diego, CA, USA). Results were considered statistically different when $p < 0.05$. For specific growth curve: DF: 26; Exact *p*-value: 0.0070/For antibiogram analysis: DF: 26 (for every antibiotic tested); Exact *p*-value: 0.0055, 0.041, 0.0003, 0.0341 (for gentamicin, oxytetracycline, streptomycin and oxolinic acid, respectively). Principal component analysis (PCA) was employed resorting to the GraphPad Prism 9 for Windows to disclose the overall network among the seven Portuguese strains of *E. amylovora* and between them and type strain LMG 2024 and reference strain CFBP 1430 regarding their phenotypic characterization.

# RESULTS

## Growth curve and specific growth rate

Every strain tested displayed similar growth curves (Fig. S1). Nevertheless, they presented statistically significant differences regarding their μ (Fig. 1), namely, strain Ea 630 had lower μ(0.206 ± 0.049 h-1) ($p < 0.05$) than strains LMG 2024, Ea 490, Ea 680, and Ea 820, whilst the remaining strains did not present significant differences. Furthermore, it was possible to observe that the strain Ea 490 had the highest μ(0.372 ± 0.072 h-1), and was significantly different ($p < 0.05$) when compared with strain Ea 630 (Dataset S1).

## *Erwinia amylovora* strains phylogeny

The dendrogram generated from the alignment of the four concatenated MLSA housekeeping genes (*gapA*, *gyrB*, *infB*, and *rpoD*) disclosed a high genetic homogeneity between the Portuguese strains, with the seven strains located in a single 100% boot-strap

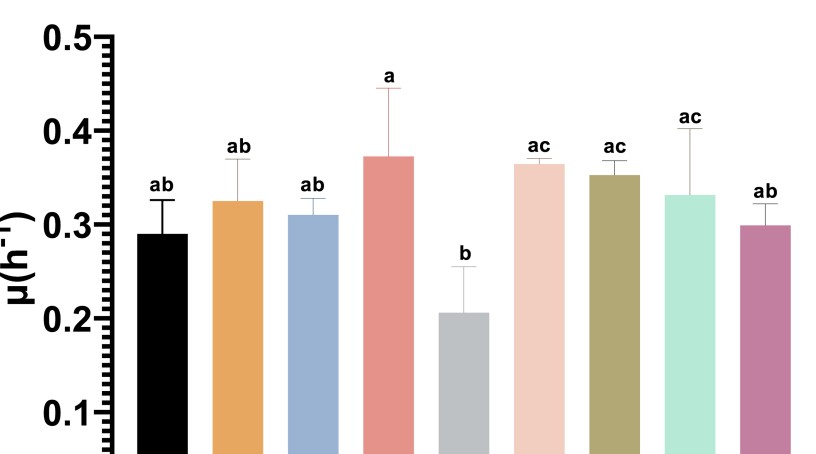

**Figure 1** **Specific growth rate of the seven *Erwinia amylovora* strains, type strain LMG 2024 and reference strain CFBP 1430.** Vertical bars: mean value with standard deviation ($n = 3$), $p < 0.05$. Different letters mean statistical significance. Statistical analysis: One-way ANOVA, followed by Tukey's.

supported clade together with the type strain LMG 2024 and the 16 reference strains (Fig. 2). Moreover, no polymorphisms were found between the seven *E. amylovora* strains and with the remaining reference strains of these bacteria. The other strains of *Erwinia* spp. genus appeared in other branches of the phylogenetic tree, with *Erwinia pyrifoliae* being the most closely related, whilst *Erwinia rhapontici* and *Erwinia bilingiae* the furthest.

## Carbon source and chemical susceptibility

The seven strains studied in this work were identified as *E. amylovora* according to the Biolog GEN III Database (version 2.8). However, these strains displayed different biochemical profiles regarding the use of the carbon sources and their chemical susceptibility (Table S4). The seven strains, together with type strain LMG 2024 and reference strain CFBP 1430 presented differences in the use of glycyl-L-proline, methyl pyruvate, L-alanine, myo-inositol, D-cellobiose, D-salicin, L-aspartic acid, citric acid, keto-glutaric acid, D-fructose-6-PO4, acetic acid, inosine, L-serine, and bromo-succinic acid. These same strains display higher differences regarding their chemical response to lincomycin, aztreonam, 4% NaCl, fusidic acid, guanidine, HCl, sodium butyrate, pH 5, 8% NaCl, D-Serine, and sodium bromate. These biochemical profiles allowed to disclose in two dendrograms (one for the carbon use and other for the chemical susceptibility) different clustering among the strains (Figs. 3 and 4). Regarding the carbon sources, three different clusters for the seven
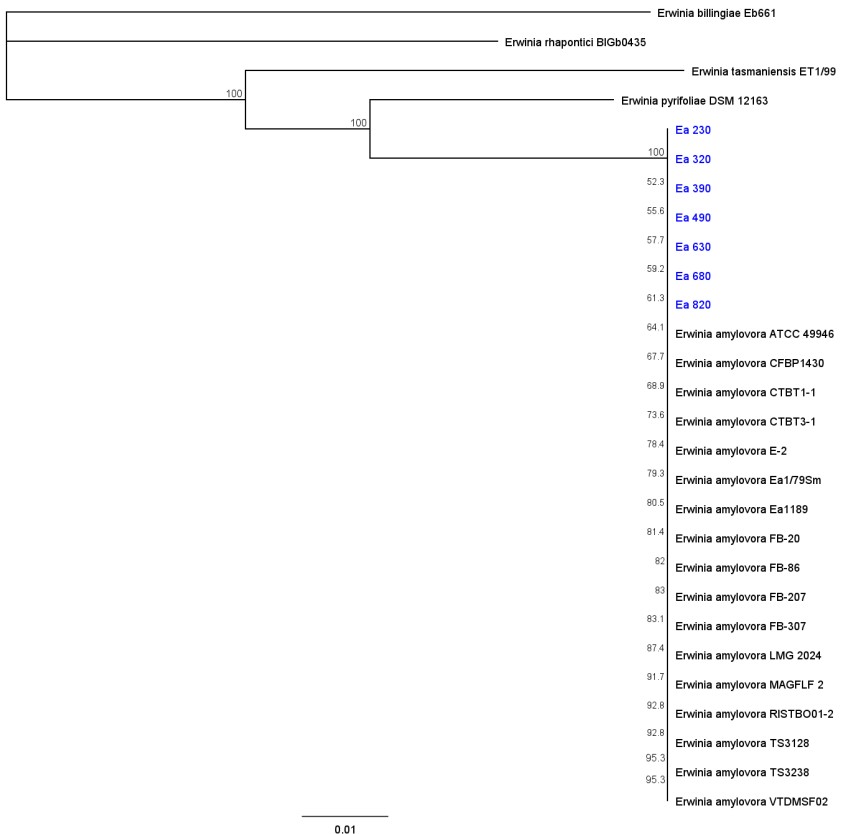

**Figure 2** **Phylogenetic tree.** Phylogenetic tree using four concatenated housekeeping genes (*gapA*, *gyrB*, *infB*, and *rpoD*) with 2,276 bp from seven Portuguese *Erwinia amylovora* strains (Ea 230, 320, 390, 490, 630, 680, and 820) (blue), available strains of *E. amylovora*, and closely related of the genus *Erwinia*. Every strain used to build the dendrogram are available in Table S1. Neighbour-joining method (bootstrap: 1,000 replicates) was used to construct the phylogenetic tree on Geneious Prime software, scale bar represents the number of the nucleotide substitutions per site.

Portuguese *E. amylovora* isolates, grouping together Ea 230 with Ea 490, Ea 320 with Ea 680, and Ea 630 with Ea 820, whilst Ea 390 was closely related to the first two groups (Fig. 3). Moreover, the dendrogram showed that the Portuguese strains were more related among them than with type strain LMG 2024 and reference strain CFBP 1430, with the last being the less related of the nine tested strains (Fig. 3).

On the other hand, the resulting dendrogram from the chemical susceptibility showed two major clusters, with the strain Ea 630 being grouped with type and reference strain, and the remaining strains in another group, with strains Ea 230 and Ea 490 paired together and the others in an ascending pattern (Fig. 4).

## Antibiotic susceptibility

To assess the susceptibility of the seven Portuguese strains of *E. amylovora*, type strain LMG 2024, and reference strain CFBP 1430 to four antibiotics that are applied to control fire blight, an antibiogram assay was carried out (Dataset S2). This assay disclosed that the strains tested against oxytetracycline and oxolinic acid did not present significant
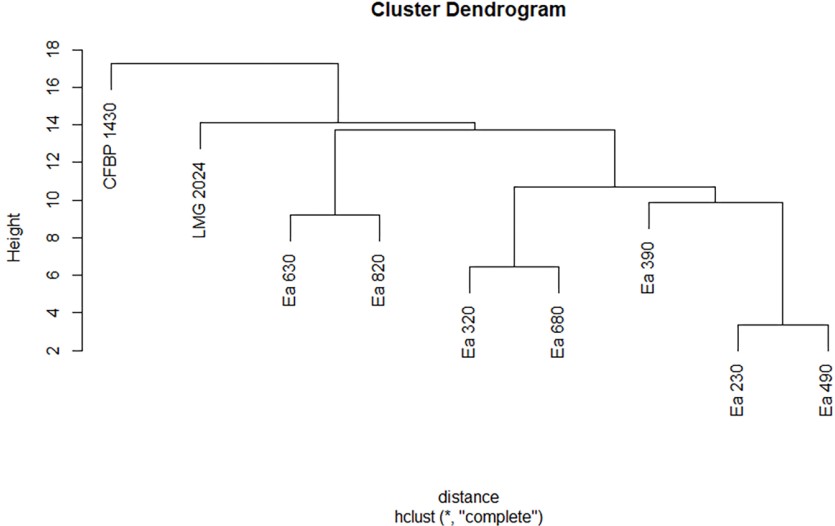

**Figure 3 Dendrogram displaying phenotypic diversity among the seven Portuguese *Erwinia amylovora* strains, type strain LMG 2024 and CFBP 1430 regarding the use of different carbon sources.** Dendrogram was built from Table S3 (black) that summarises the biolog pattern for each strain tested.

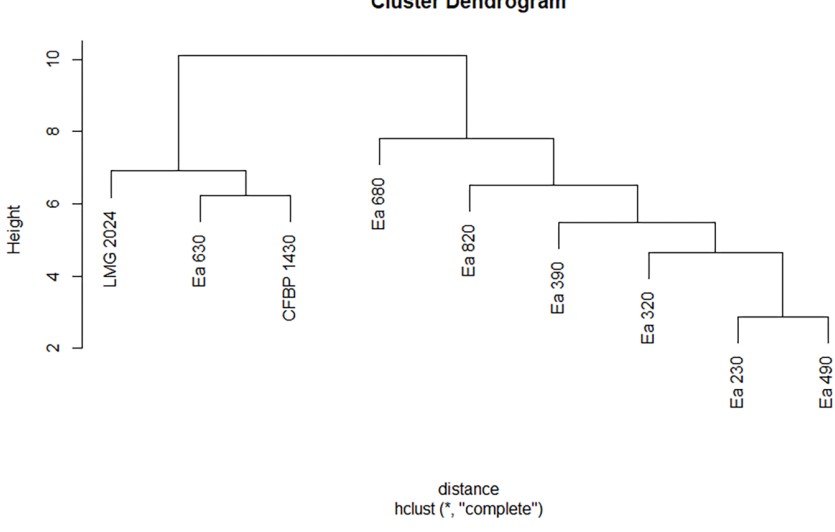

**Figure 4 Dendrogram displaying phenotypic diversity among the seven Portuguese *Erwinia amylovora* strains, type strain LMG 2024 and CFBP 1430 regarding their chemical susceptibility.** Dendrogram was built from Table S3 (red) that summarises the biolog pattern for each strain tested.

differences ($p > 0.05$) in the susceptibilities among them (Figs. 5B and 5D). Nevertheless, the strains showed a trend regarding their halo diameter, with strains Ea 390 and Ea 680 being more susceptible to oxytetracycline, whilst the strains Ea 490 and Ea 630 were less susceptible, presenting a similar halo to type strain LMG 2024 (Fig. 5B). For oxolinic acid,

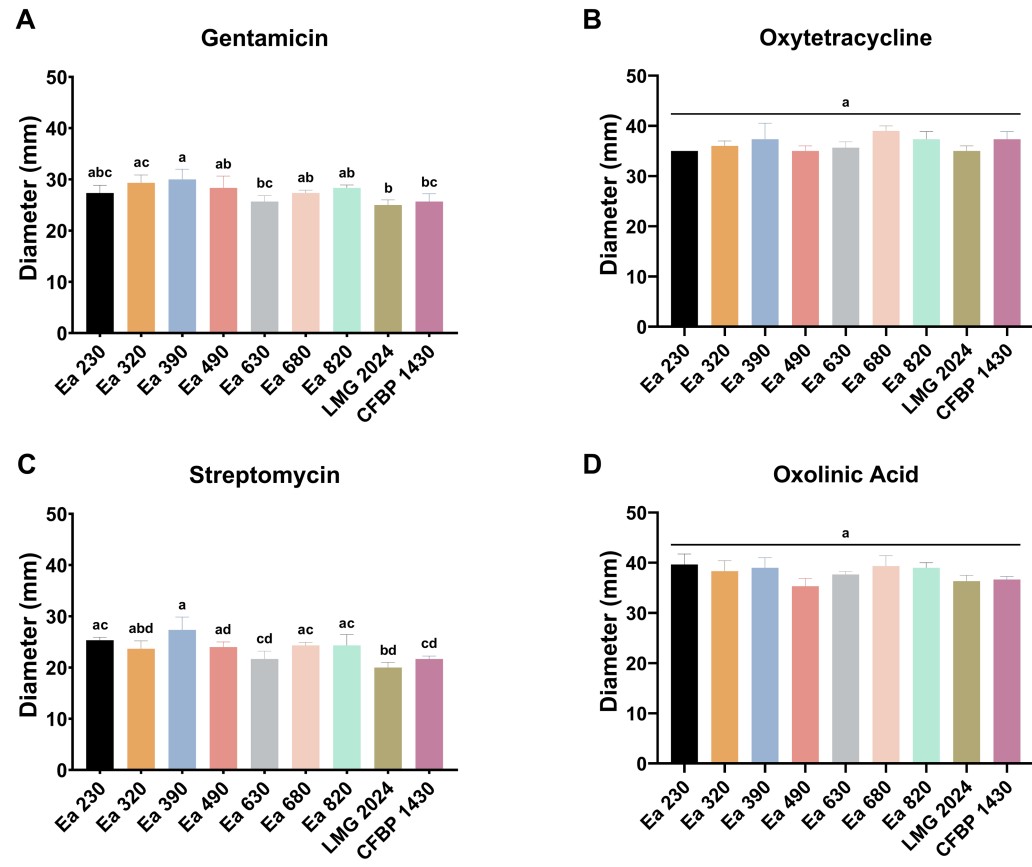

**Figure 5** **Antibiogram results of the seven Portuguese *Erwinia amylovora* strains, type strain LMG 2024, and reference strain CFBP 1430.** (A) Gentamicin, (B) oxytetraclycline, (C) streptomycin and (D) oxolinic acid. Vertical bars: mean value with standard deviation ($n = 3$), $p < 0.05$. Different letters mean statistical significance. Statistical analysis: One-way ANOVA, followed by Tukey's.

the same trend was observed with strains Ea 490 and Ea 630 being less susceptible to it and strains Ea 230, Ea 390, and Ea 680 being more susceptible (Fig. 5D). The gentamicin susceptibility presented several statistically significant differences ($p < 0.05$), namely, strain Ea 390 was the most susceptible to the antibiotic, whilst strain Ea 630 was the less susceptible of the Portuguese strains, while the type strain LMG 2024 was the less susceptible of all the strains tested. Moreover, the different susceptibility of the most and the least susceptible strains (Ea 630, LMG 2024, and CFBP 1430) was statistically significant. Streptomycin susceptibility profile of the strains presented statistically significant differences ($p < 0.05$), with the same behavior observed for the strains Ea 390 (most susceptible) and strains Ea 630, LMG 2024, and CFBP 1430 (least susceptible). Additionally, it was possible to differentiate strains Ea 230, Ea 680, and Ea 820 from the strain LMG 2024, with the first group being more susceptible than the type strain. Further analysis of the antibiogram results shows that for every strain tested, the antibiotic Streptomycin is where the strains presented less susceptibility and the oxolinic acid where more susceptibility was shown (Fig. 5).

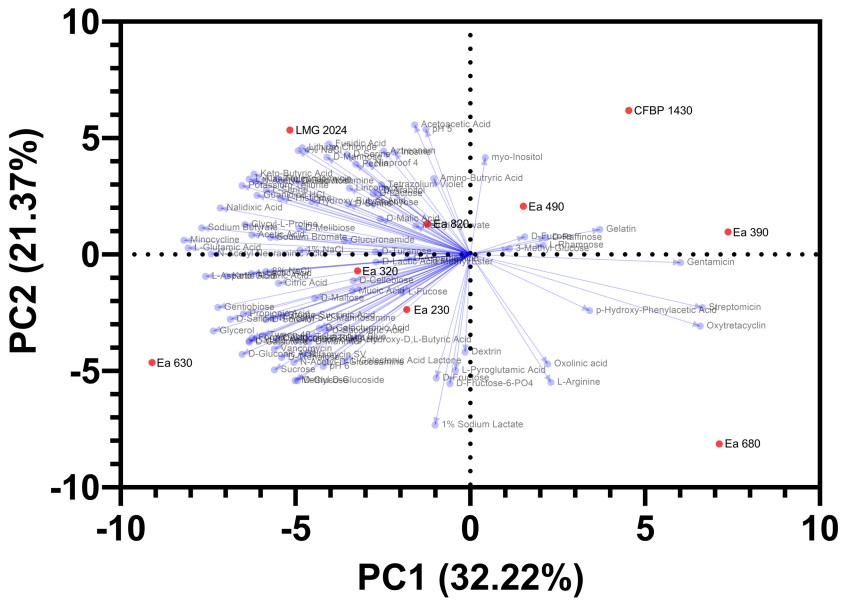

**Figure 6** Principal component analysis, regarding Biolog and antibiogram characterization, of the seven Portuguese *Erwinia amylovora* strains, type strain LMG 2024 and reference strain CFBP 1430.

## Multivariate analysis

PCA showed a clear separation among the seven strains of *E. amylovora* collected in Portugal, the type strain LMG 2024, and the reference strain CFBP 1430, regarding the phenotypic characterization (Fig. 6). PC1 explained 32.2% of the variance and PC2 21.4% of the variance. In the PCA analysis, two distinct groups are identified: (1) the upper right quadrant that groups the reference strain CFBP 1430 and the strains Ea 490 and Ea 390; and (2) the strains Ea 230, Ea 320, and Ea 820, that group together in the center of the PCA. Besides that, two Portuguese strains are clear outliers of the population tested, with strain Ea 630 being positioned in the far left of the lower-left quadrant and the strain Ea 680 positioning herself in the far right of the lower right quadrant. Type strain LMG 2024 also isolates itself, being positioned in the upper left quadrant. This analysis allows identifying, and grouping, the strains that possess similar phenotypic traits evaluated (two distinct groups), and the strains that differentiate themselves from the remaining population.

## DISCUSSION

Fire blight remains a major problem for apple and pear production in several European countries, including Portugal (*EPPO, 2018*), economically affecting several producers and important chain-of-value fruits, such as *P. communis* cv. 'Rocha'. Nevertheless, the first molecular characterization of strains related to Portuguese outbreaks was only disclosed recently, showing a highly clonal population of *E. amylovora* (*Mendes et al., 2021a*). However, phenotypic diversity was found among the strains regarding their virulence, and susceptibility to antimicrobial peptides, suggesting some diversity among the strains. This work further characterized this population, using the most representative

strains of the previous study, resorting to other molecular genotyping, and in diverse phenotypic characteristics.

MLSA genotyping re-confirmed the genetic homogeneous population of *E. amylovora* in Portugal, previously characterized by CRISPR genotyping (*Mendes et al., 2021a*), and is in line with a study from Kyrgyzstan, that also presented a clonal population of this bacterium resorting to MLSA (*Doolotkeldieva et al., 2019*). Considering that these strains are representative of several outbreaks that occurred between 2010 and 2017 in different areas of Portugal (and from different hosts), these results also show a high genetic homogeneity throughout the years. Moreover, this analysis included other *E. amylovora* strains that were collected in other countries, namely South Korea, Belarus, France, Germany, the United Kingdom, and USA, and they were all clustered together. This may be due to a high specialization from this bacterium to the same hosts since the 1980s because of pome fruit breeding strategy that favors high-valued varieties, that are often susceptible to fire blight (*Mann et al., 2013*; *Piqué et al., 2015*). Another factor that can explain this global low variability is the fact that these strains were collected from *P. communis* and *M. domestica*. These two host species are from the Amygdaloideae sub-family, which have been stated in previous studies that may provide much more homogeneous *E. amylovora* strains than the Rubus-infecting strains (*Rezzonico, Smits & Duff, 2011*; *McGhee & Sundin, 2012*; *Mann et al., 2013*; *Bühlmann et al., 2014*; *Piqué et al., 2015*; *Zhao et al., 2019*). Additionally, the studies of *Zeng et al. (2018)* and *Parcey et al. (2020)* have demonstrated that strains collected from Europe are grouped in the widely-prevalent clade that has originated in North America and disseminated worldwide.

Phenotypic characteristics of bacteria disclose traits that allowed them to survive in the environment, such as, their virulence fitness, resistance to chemical compounds, among others, making their characterization important to understand population behavior, evolution, and dynamics. Bacterial growth is a core phenotypic trait, capable to rapidly distinguish strains. It is known that several factors can affect microbial growth, such as pH, temperature, and nutrients. Growth curve results showed that every strain tested was able to start their exponential growth rapidly (∼3 h), which is expectable, since it is known that *E. amylovora* optimal growth occurs around 28 °C (*EPPO, 2013*; *Santander & Biosca, 2017*). However, their specific growth rate ($\mu$) varied between the strains, with the most significant difference being visible for the strain Ea 630 that displayed the lowest $\mu$ than the rest of the studied strains. This indicates that these strains are different regarding their growth kinetics, which may result in differences regarding their virulence since it is known that $\mu$ can regulate the production of exopolysaccharides (EPS), such as biofilms and amylovoran, that are essential for *E. amylovora* virulence (*Piqué et al., 2015*). Indeed, a previous study showed that lower $\mu$ led to higher EPS production (*Evans, Brown & Gilbert, 1994*), thus to higher virulence by the bacterium. This report is corroborated by the previous results obtained with these strains, where one of the most virulent strains was Ea 630 and one that was less virulent was Ea 680 (*Mendes et al., 2021a*). Now it was possible to understand that Ea 630 had a lower $\mu$ than Ea 680.

Biolog systems have been used to identify and characterize different plant pathogenic bacteria (*LeBlanc et al., 2017*; *Flores et al., 2018*) and can distinguish different *Erwinia* spp.,

namely *E. amylovora*, *Erwinia persicina*, and *E. rhapontici*. Some previous studies have used Biolog for the identification and discrimination of *E. amylovora* strains (*Atanasova et al., 2007*; *Donat et al., 2007*; *Constantinescu et al., 2011*; *Ivanović et al., 2012*; *Marco-Noales et al., 2017*). In the seven Portuguese *E. amylovora* strains, type strain LMG 2024, and reference strain CFBP 1430 it was possible to observe that they metabolize differently 14 carbon sources and respond differently to 12 chemical compounds, which allowed to clearly discriminate the strains isolated in Portugal from the type and reference strain. The differences between the Portuguese strains and the reference and type strains consisted in their inability to metabolize glycyl-L-proline, methyl pyruvate, L-alanine, myo-inositol, D-salicin, keto-glutaric acid, D-fructose-6-PO4, acetic acid, among others, to the same extent. These results show a diversity in the carbon usage between the Portuguese strains, clearly separating them as it is possible to observe in the dendrogram, as has been demonstrated before for other strains in a previous study (*Atanasova et al., 2007*). Regarding their chemical susceptibility, Biolog showed that all Portuguese strains are resistant to troleandomycin, vancomycin, and rifamycin SV antibiotics and susceptible to nalidixic acid and minocycline, with strain Ea 680 being the only susceptible to lincomycin. Moreover, strains Ea 630 and Ea 680 were susceptible to aztreonam. Besides that, diversity regarding sensitivity to other factors was observed between the Portuguese strains and the type and reference strains. This may indicate that environmental adaptations may have occurred in these strains, which may represent processes of local evolution favoring *E. amylovora* variability. Regardless the differences regarding carbon source preferences and chemical susceptibility, the dendograms do not support an evident linkage between these phenotypic traits (carbon sources and antibiotic susceptibility) for the studied strains.

Even though the use of antibiotics is forbidden in the EU, with few regulated exceptions (*Sundin & Wang, 2018*), *E. amylovora* antibiotic resistance strains are increasing (*Tancos & Cox, 2016*; *Dafny-Yelin et al., 2020*), which calls for the need to determine the susceptibility of the Portuguese strains to antibiotics authorized in some countries to control fire blight (*Sundin & Wang, 2018*). The oxytetracycline and oxolinic antibiograms recorded for all the strains tested did not indicate significant differences, but it was possible to observe different susceptibilities among strains, with a halo diameter superior to 30 mm, which was previously found for oxytetracycline (*Atanasova et al., 2007*), demonstrating a high susceptibility for these antibiotics. On the other hand, for gentamicin and streptomycin, it was possible to distinguish significant differences between the strains regarding their susceptibility to these antibiotics. For both antibiotics, the strain Ea 630 was the less susceptible of the *E. amylovora* Portuguese strains, whilst the strain Ea 390 displayed the highest susceptibility to both antibiotics, which might result from different environmental pressures such as prior exposure to antibiotics, especially concerning strain Ea 630 isolated in 2015. For both antibiotics, the Portuguese strains presented higher susceptibility than type strain LMG 2024 and reference strain CFBP 1430, presenting halo diameters ranging between 25 and 30 mm for gentamicin and 22 to 28 mm for streptomycin, which was previously found in another study (*Atanasova et al., 2007*). This may indicate that the introduction of these strains was originated from countries where *E. amylovora* did not develop high resistance to these antibiotics. Overall, every *E. amylovora* strain isolated in

Portugal did not present any resistance to the antibiotics tested, as it has been previously found in strains from Spain (*Donat et al., 2005*), with streptomycin and oxolinic acid resistant strains being discovered until now in the USA, New Zealand and Israel (*Stockwell & Duffy, 2012*; *Sundin & Wang, 2018*).

Despite the molecular/genetic approaches regarding specific genome regions of the strains (CRISPR and MLSA regions) support a highly clonal population for the studied *E. amylovora* strains, taking into account the phenotypic profiling carried out and filtered through a PCA, the data points towards two distinct groups of *E. amylovora* within the studied strains and shows two clear outliers. Although, no immediate correlation could be inferred for the origin of these strains, phenotypic heterogeneity was discerned through several phenotypic traits (growth rates, metabolism of carbon sources and chemical susceptibility), which may support the hypothesis that the fire blight outbreaks occurring in Portugal between 2010 and 2017 may have had different origins. Moreover, strains Ea 630 and Ea 680 are worthy of further attention, since they were both more and less virulent in a previous study and the last one displayed a new CRISPR profile (*Mendes et al., 2021a*). Phenotypic variability among a population of *E. amylovora* isolates from Georgia with two CRISPR profiles has been disclosed, suggesting that multiple entry events may have occurred in that country (*Gaganidze et al., 2021*), which is similar to the findings presented here. Regarding the clonality demonstrated previously through CRISPR (*Mendes et al., 2021a*) and here corroborated for the studied strains by MLSA, future research should extend the assessment of clonality to a representative number of populations by microsatellites typing, resorting to VNTR or MLVA, or through comparative genomics as explored in previous works on *E. amylovora* (*Alnaasan et al., 2017*; *Zeng et al., 2018*; *Parcey et al., 2020*; *Tafifet et al., 2020*), to determine the existing haplotypes allowing to infer source tracking. In this work, strains Ea 630 and 680 demonstrated different antibiotic susceptibility, with the first being more resistant than the other, which could indicate that these strains have been exposed to different environmental pressures, making them candidates for different sources of *E. amylovora* in Portugal and suitable candidates for epidemiological studies.

## CONCLUSIONS

This work further confirmed, through MLSA, previous molecular results disclosing a genetic clonal population of *E. amylovora* in Portugal, supporting the high clonality found for this species. Nevertheless, variability among the studied *E. amylovora* isolates was particularly observed regarding their specific growth rate ($\mu$), carbon source use and chemical sensitivity, and susceptibility to antibiotics, which could be explained by several factors, from plasmid content to genomic islands, among others that could have a role in this diversity alongside other factors.

Finally, multivariate analysis, supported the phenotypic diversity among the *E. amylovora* Portuguese strains, which shows two *E. amylovora* strains different among the population, namely Ea 630 and Ea 680, that could be due to the chemical sensitivity and antibiotic susceptibility as seen in the results, ultimately supporting the hypothesis of fire blight

outbreaks in Portugal resulting from independent events. This study can open the possibility of future studies down the line regarding this heterogeneity, as a key factor for different sources and/or evolution/adaptation of this pathogen against control managements and environmental adaptation.

Overall, besides confirming the clonality of the studied *E. amylovora* isolates, this study details the first comprehensive phenotypic characterization of *E. amylovora* strains isolated from fire blight outbreaks in Portugal, which provides further insight into possible strain-specific adaptations to environmental stresses, contributing to a better understanding of *E. amylovora* diversity and its epidemiology.

### Funding

This work received financial support by public funds by Ministério da Ciência e Tecnologia –Fundação para a Ciência e a Tecnologia through the project financed with reference UIDB/50006/2020 | UIDP/50006/2020, and Rafael J. Mendes received support from Ministério da Ciência e Tecnologia –Fundação para a Ciência e a Tecnologia (grant number SFRH/BD/133519/2017). The funders had no role in study design, data collection and analysis, decision to publish, or preparation of the manuscript.

### Grant Disclosures

The following grant information was disclosed by the authors:
Ministério da Ciência e Tecnologia –Fundação para a Ciência e a Tecnologia: UIDB/50006/2020 | UIDP/50006/2020, SFRH/BD/133519/2017.

### Competing Interests

The authors declare there are no competing interests.

### Author Contributions

- Rafael J. Mendes conceived and designed the experiments, performed the experiments, analyzed the data, prepared figures and/or tables, authored or reviewed drafts of the article, and approved the final draft.
- Conceição Amaro performed the experiments, authored or reviewed drafts of the article, and approved the final draft.
- João Pedro Luz conceived and designed the experiments, authored or reviewed drafts of the article, and approved the final draft.
- Fernando Tavares conceived and designed the experiments, analyzed the data, authored or reviewed drafts of the article, and approved the final draft.
- Conceição Santos conceived and designed the experiments, analyzed the data, authored or reviewed drafts of the article, and approved the final draft.

### Data Availability

The raw data used for specific growth rate and antibiogram results (Figs. 1 and 5) is available in the Supplementary Files.

The sequences are available in GenBank: MW647223, MW647224, MW647225, MW647226, MW647227, MW647228, MW647229, MW647230, MW647231, MW647232, MW647233, MW647234, MW647235, MW647236, MW647237, MW647238, MW647239, MW647240, MW647241, MW647242, MW647243, MW647244, MW647245, MW647246, MW647247, MW647248, MW647249, MW647250.

## Supplemental Information

Supplemental information for this article can be found online at http://dx.doi.org/10.7717/peerj.13695#supplemental-information.

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
