# Peer review of "Variability within a clonal population of Erwinia amylovora disclosed by phenotypic analysis"

_PeerJ, doi:10.7717/peerj.13695_

## Round 0.1 · original submission · Minor Revisions

Dear Authors, the three reviewers agree that this is a great work that should be published. there are only small details that need to be fixed. Congratulations.

Reviewer 1 ·

Basic reporting

1. Manuscript writing is clear using unambiguous professional english.
2. Literature references of sufficient field background is, however, not exhaustive.
3. Article structure is professional, figures and tables are sufficient to support results.

Experimental design

Authors of the manuscript aimed to further disclose the genetic characterization and unveil the
phenotypic diversity of this E. amylovora population, resorting to MLSA, growth kinetics, biochemical characterization, and antibiotic susceptibility. This research question is not only well defined but also relevant and meaningful towards providing a more integrated approach in epidemiological surveillance of fireblight in Portugal.

However, the manuscript does not clearly state criteria used to select the chosen phenotypic evaluation (growth kinetics, carbon usage and antibiotic resistance).

For instance, carbon source usage is not directly related to virulence (the variable trait shown among isolates).

On the other hand, growth kinetics as a phenotypic trait to profile E. amylovora diversity, could be relevant if psychrotrophic adaptations were considered (Santander RD, Biosca EG. Erwinia amylovora psychrotrophic adaptations: evidence of pathogenic potential and survival at temperate and low environmental temperatures. PeerJ. 2017 Oct 26;5:e3931. doi: 10.7717/peerj.3931).

Additionally, "Biochemical characterization" as a title for a phenotypic evaluation might be misleading since the BIOLOG GEN III microplate analysis, while robust, is based only on carbon sources usage. This reviewer opinion is that authors should avoid the term "biochemical characterization" and instead use a more specific term (i.e. carbon source usage, or similar).

Carbon-source analysis does provide useful information for identification and fingerprinting at species level, but strains sample size is showing not to be enough to provide profiling data for Erwinia virulent isolates. Moreover, if molecular epidemiological characterization had shown clonality among isolates displaying diversity in virulence, this reviewer finds it hard to understand why phenotypic characterization based on virulence determinants (i.e. biofilm, motility, amylovoran production, siderophores) was not considered in the evaluation.

Attempting to find traits to further high-throuput profile E. amylovora highly clonal isolates, is of urgent relevance in the light of frequent pome-trees fire blight outbreaks. If this is the main driver underlying phenotypic analysis selection, it should be clearly stated early in a more comprehensive version of the manuscript.

Validity of the findings

Individually, experiments provide robust, statistically sound, and controlled data.

However, the manuscript lacks support in the selection and extent of phenotypic analyses. An improved manuscript should provide a benchmark against appropriate references or a suitable justification for the criteria behind growth kinetics and carbon source analysis in particular for these isolates.

Conclusions state that further phenotypic assessment of E. amylovora isolates is needed (due to isolates clonality); nevertheless, authors conclusions are vague and require a deeper analysis of provided results to fully reach their stated goal: "to assess growth kinetics, biochemical variability (carbon usage), antibiotics susceptibility, and phylogenetic markers, as features capable to discriminate E. amylovora strains within a genetically clonal population".

To link research question to conclusions, analysis could deepen on the phenotypic profiling that results display; i.e. are there phenotypes associated to phylogenetic markers? Does PCA (carbon usage) grouping relate to dendograms or phylogeny?

An improved manuscript containing a more coherent and straightforward conclusion, would explain the manuscript main contribution to knowledge in the area.

Additional comments

Minor corrections:
-- Line 279: PCA showed a clear separation between the seven strains of E. amylovora collected in Portugal, the type strain LMG 2024, and the reference strain CFBP 1430, regarding the
281 phenotypic characterization (Fig. 6).
R- is the word "between" correctly used in this sentence? aren´t authors comparing among groups?

·

Basic reporting

The authors report the variability within clonal population of Erwinia amylovora disclosed by phenotypic analysis like biochemical characterization and antibiotics susceptability.
The manuscript is really well written, in professional english. The background information and references provided are sufficient to clearly understand the topic and to perfectly follow the work.
Figures and tables well reports the experimental results.

Experimental design

The experimental design is straightforward and well planned. Starting from the basic phylogenetic analysis on housekeeping genes, the authors focused on phenotipic characterization of the clonal population, based on biochemical characterization and antibiotics susceptability. The results were statistically analyzed with the standard methods (Anova, Tukey's, PCA).

Validity of the findings

The results give new insights on the clonalality of Erwinia amylovora. Regardless the low genetic variability, the authors were able to identify some phenotipic differences.
The authors also conclude and state that the results may lead future analysis.

Additional comments

I found the manuscript really interesting and well written and I suggest to accept it in the present form.
However, I found some minor mistakes in the text that I suggest to correct before publication:

Lane 90: McManus et al. 2020 is reported in the References as McManus et al. 2002, which is the correct ublication year. Please correct the text.

Lane 380: remove the - in intro-duction. (Maybe the formatting changed during the pdf creation).

Lane 429-430: as above, I see too many spaces in the senteces.

Lane 462: I do not find Donat et al. 2005 in the text. Please correct.

Lane 591: Prunus domestica goes in italic.

Figure 1 and Figure 5: Why are you using the same colors for Ea230 and Ea820? Or same orange and light blu for Ea320-LMG2024 and Ea390-CFBP1430? Do they represent same cluster? If yes, better state in the figure captions or use different colors for each of the strain.

The authors state that future analysis will probably performed based on MLVA analysis. What about whole genome sequencing? It is known that MLSA may not be sufficient to discern highly clonal population and MLVA may be labor and time consuming. Nowadays WGS is quite cheap and easy to perfomed for bacteria. Think about it.

Reviewer 3 ·

Basic reporting

The article is written in clear unambiguous professional English.

Literature references and context is provided.

The article has a professional structure: Figures and tables are relevant to the content. Raw data are shared.

The article represents an appropriate "unit of publication".

Experimental design

The article contains original research within the scope of the journal.

Research questions are well-defined and meaningful. The aims of the study are clearly stated.

The investigations were performed at at a high technical standard.

The methods are described with sufficient detail to replicate the study.

Validity of the findings

All raw data have been provided, they are statistically sound.

Conclusions are well stated and linked to the original research question.

Additional comments

The article covers an important topic, i.e. the diversity within E. amylovora populations in Portugal. In general, the article is well-written and the methods used appropriate to solve the research question. I only have some minor comments on the Discussion section:

-L402: Why is whole genome sequencing not discussed as a method to uncover diversity within E. amylovora populations? Could this be added as a further research topic? Genome sequencing has been used previously in E. amylovora populations ( Sing&Kahn 2019, https://www.nature.com/articles/s41598-019-50589-z, and Parcey et al, 2020 https://www.sciencedirect.com/science/article/pii/S0888754320301117).

-L413: minor format error (E. amylovora not in italics)

I also have one comment regarding the Figure legends:

-Fig. 1 and Fig. 5: it would be helpful if the statistical test used to infer significant differences would also be stated in the legend (it is a bit bothersome to search again for it in the main text).

---

## Round 0.2 · accepted · Accept

I believe all the issues have been solved and the manuscript is ready.